# Unravelling the Link between Psychological Distress and Liver Disease: Insights from an Anxiety-like Rat Model and Metabolomics Analysis

**DOI:** 10.3390/ijms241713356

**Published:** 2023-08-29

**Authors:** Binjie Liu, Shanshan Zhang, Lizhu Sun, Lan Huang, Rong Zhang, Zhongqiu Liu, Lin An

**Affiliations:** Guangdong-Hong Kong-Macau Joint Lab on Chinese Medicine and Immune Disease Research, Guangdong Provincial Key Laboratory of Translational Cancer Research of Chinese Medicines, Joint International Research Laboratory of Translational Cancer Research of Chinese Medicines, International Institute for Translational Chinese Medicine, School of Pharmaceutical Sciences, Guangzhou University of Chinese Medicine, Guangzhou 510006, China; 20201110556@stu.gzucm.edu.cn (B.L.); 13959764313@139.com (S.Z.); 20211110086@stu.gzucm.edu.cn (L.S.); hollieshuang0925@163.com (L.H.); zhangrong@gzucm.edu.cn (R.Z.)

**Keywords:** anxiety-like behavior, EGFR, metabolomics, HPA axis, inflammation

## Abstract

Psychological distress is associated with an increase in liver disease mortality. This association highlights the close relationship between psychological and physical health. The underlying mechanism of this association needs to be elucidated. In this study, a rat model of anxiety was developed via compound stress. Changes in the HPA axis and inflammatory factors in the brains of the rats were evaluated for behavioral tests and liver function, respectively. The liver metabolic profiles of the rats were characterized through liquid chromatography–mass spectrometry (LC-MS). Differential metabolites were screened based on the conditions of *p* < 0.05 and VIP > 1. A pathway enrichment analysis was performed on the metabolomics data using the Ingenuity Pathway Analysis (IPA). Immunofluorescence (IF), immunohistochemistry (IHC), and Western blotting assays were performed to examine the expression of the screened target epidermal growth factor receptor (EGFR) and to elucidate the pathway associated with the mechanism. The results showed the impairment of liver function among the rats in an anxiety-like state. Additionally, 61 differential metabolites in the control and anxiety groups were screened using metabolomics (*p* < 0.05, VIP > 1). The results of the IPA analysis showed that the key target was EGFR. We also found that an anxiety-like state in rats may cause liver injury through the EFGR/PI3K/AKT/NF-κB pathway, which can lead to the production of inflammatory factors in the liver. Our results revealed a mechanism by which anxiety-like behavior leads to liver damage in rats. The findings of this study provided new insights into the deleterious effects of psychological problems on physical health.

## 1. Introduction

Mental health issues are a serious problem in modern society. The number of cases of psychological problems increased during the COVID-19 pandemic. In the first year of the pandemic, depression and anxiety increased by more than 25% [1,2]. Anxiety is a serious condition prevalent in Europe and the United States [3,4,5]. Considering that the lifetime prevalence of anxiety disorders is currently very high (13.6–28.8%), and the age of onset of mental illness is only 11 years, this problem is a serious concern. People with anxiety disorders often suffer from other comorbidities, including not only mental illnesses, such as depression [4,5,6,7], but also medical conditions, including functional gastrointestinal disorders, asthma, cardiovascular diseases, cancer, chronic pain, high blood pressure, and migraine [8,9,10].

An increasing number of articles have approached the link between psychosocial stress and the negative evolution of hepatic diseases [11]. Psychological distress is a predictor of liver disease, while liver injury may result in deeper psychological distress and lead to a vicious cycle. Once an individual is subjected to a stressor, specific pathways within the brain lead to the activation of the hypothalamic-pituitary-adrenal (HPA) axis as well as the central sympathetic outflow. Stress-related dysregulation of the HPA axis may lead to the secretion of proinflammatory substances from the liver, such as tumor necrosis factor (TNF) and interleukin-6 (IL-6), which in turn can cause liver inflammation. Hence, anxiety disorders are a major burden and have social and financial repercussions [4,6]. However, only a few studies have investigated liver damage caused by psychological distress, and the mechanism is not clear. Hence, further studies are necessary for the efficacious diagnosis and treatment of anxiety disorders. Additionally, information on the pathological process, etiology, and causative factors of anxiety disorders is also extremely limited [12,13].

Metabolomics is an interdisciplinary field that has emerged in the post-genome era. By conducting mass spectrometry, complex data can be analyzed in multiple dimensions, and important information can be obtained, which can greatly improve the accuracy of the analysis of the metabolic properties at various physiological and pathological stages in the human body [13,14,15,16,17]. Therefore, the relevant differential metabolites and related targets can be determined using the metabolomics approach for accurately diagnosing anxiety disorders. Although several studies have reported that psychological problems affect liver health, the molecular mechanisms involved in this process are only partly understood.

To investigate these issues, we constructed a compound stress rat model and found that the rats exhibited anxiety-like behavior, along with hyperfunction of the HPA axis and neuroinflammation. We also found that rats showing anxiety-like behavior developed liver injury, which was then investigated via untargeted metabolomics of the liver tissue using liquid-chromatography-quadrupole time-of-flight mass spectrometry (LC-QTOF-MS). We found that anxiety-like behavior caused liver injury to some extent, and it was closely associated with the epidermal growth factor receptor (EGFR)-related pathway. These findings suggested that mental health is important for physical harmony. The workflow of this study is summarized in Figure 1.

## 2. Results

### 2.1. Composite Stress-Induced Anxiety-like Behaviors in Rats

We investigated the role of stress-related behavioral modifications using a model of anxiety generated by tail pinching and persistent restraint for 14 consecutive days. The anxiety index of stress-induced rats was calculated using activity tracking during the elevated cross-maze test (EPM). The results showed that the arms of the stressed group were mostly closed (Figure 2A). The anxiety index was considerably higher in the anxiety group than that in the control group (Figure 2C). 

The OFT was performed to observe anxiety-like behavior. The rats in the anxiety group moved around and did not investigate the middle area (Figure 2B), unlike those in the control group. Although the total distance traveled by the stress-induced rats in the OFT was not significantly different from that traveled by the rats in the control group, the stressed rats spent significantly less time looking in the middle area (Figure 2D).

The FST was performed to assess desperation-like behavior, and the sugar water preference experiment was performed to assess pleasure deficits. The immobility period of the rats in the anxiety group, as determined by the FST, was significantly shorter than that of the rats in the control group, which indicated that the rats did not exhibit desperation-like behavior (Figure 2E). The performance of the rats in the anxiety group was significantly poorer than that of the rats in the control group on the sucrose preference test (Figure 2F). Our results showed that a credible rat model of anxiety behavior was established, and these rats were used to conduct the remaining experiments. 

The HPA axis is central to homeostasis, stress responses, energy metabolism, and neuropsychiatric function [18]. In our study, the HPA axis becomes active after an animal experiences stress. We evaluated the ACTH and CRH levels in the blood and brain of the rats, and the CORT levels in their blood. The ACTH and CRH levels increased significantly in the serum and brain of the rats in the anxiety group (Figure 2H,I), and the serum CORT levels were also significantly higher in these rats (Figure 2J). We also recorded a significant increase in inflammatory factors in the rat brain in the anxiety group (Figure 2K,L), suggesting the hyperfunction of the HPA axis and neuroinflammation in the rats after stress.

### 2.2. Anxiety-like Behavior Aggravated Liver Injury

To evaluate the anxiety-induced hepatic injury, physical indicators and liver function were sequentially analyzed. We found that the weight of stressed rats was significantly lower than that of the rats in the control group (Figure 3A). Also, the liver coefficient was upregulated, suggesting potential hepatic damage (Figure 3B). The liver function indices showed that alkaline phosphatase (ALP), alanine aminotransferase (ALT), and aspartate aminotransferase (AST) levels were significantly higher in the serum of the rats in the anxiety group (Figure 3C,D). 

When hepatocytes are severely damaged, enzymes from their cytoplasm and mitochondria are released into the blood, resulting in a greater increase in serum AST than ALT [19]. The occurrence of chronic active hepatitis is also often higher than normal. The ratio of AST to ALT was significantly higher in the anxiety group. In clinical practice, the AST/ALT ratio is often used to identify hepatocyte injury, and it can be used as an index to assess the prognosis of liver disease. 

We also examined the oxidative stress indicators in rat liver. It is showed that both superoxide dismutase (SOD) and malonaldehyde (MDA) had remarkable alternations in the anxiety group, indicating that redox homeostasis was disrupted in the livers of the rats with anxiety-like behavior (Figure 3E,F). Further, histopathological changes were evaluated since anxiety may influence liver function. The H&E-stained liver sections obtained from the rats revealed significant inflammatory infiltrates and partial fat vacuoles in rats exhibiting anxiety-like behavior. The inflammatory infiltration areas (as shown by the arrows in Figure 3G) of six rats were significantly higher in the rats of the model group (Figure 3G). These findings suggested that anxiety-like behavior in rats may lead to liver injury. To further examine how metabolites change in the rat liver, we used a metabolomics approach.

### 2.3. Hepatic Metabolomic Analysis of Rats with Anxiety-like Behavior

To comprehensively assess the stability and repeatability of the analysis system, we examined the internal standard peak regions of many sample batches. Irrespective of whether the mass spectrum was in the positive or negative ion mode, the QC samples from different injection batches were within the range (Appendix A), which indicated the high stability and repeatability of the analytical system. The QC samples were used to assess the accuracy and reproducibility of the liver metabolomic analysis.

The results of the PCA (Appendix A) showed that the categories anxiety (A) and control (C) could be separated to different degrees in both the positive and negative ion modes. The OPLS-DA model was used to re-analyze the internal standard normalized data to emphasize the differences between groups A and C. In the positive and negative ion modes, the results indicated substantial changes in the metabolic parameters between the anxiety group and the control group (Figure 4A). However, overfitting verification was required to ensure the validity and accuracy of the OPLS-DA model. After 200 cross-validations, the results (Appendix A) showed no overfitting in the model. 

First, we screened the metabolites to gain a better understanding of the etiology of stress-induced liver injury. The metabolites with VIP > 1.0 were considered to be crucial for group separation when building the OPLS-DA model, and *p* < 0.05 was considered to be statistically significant. After removing exogenous metabolites from the HMDB database, 61 different metabolites were identified (Appendix A). Among them, 14 differential metabolites were upregulated, and 47 differential metabolites were downregulated (Figure 4B). 

The pathway enrichment analysis of the differential metabolites showed that the metabolites were primarily enriched in riboflavin metabolism, pantothenate, Coenzyme A production, arginine and proline metabolism, and other metabolic pathways (Figure 4C) (Appendix A). Next, we performed the IPA to investigate the different metabolites and found that they were mostly associated with EGFR, VEGF, TNF, and other targets (Figure 4D). 

These findings suggested that changes in liver metabolites in rats with anxiety-like behavior may lead to alterations in the potential target EGFR, and they laid the foundation for our subsequent mechanistic study.

### 2.4. Anxiety-like Behavior Aggravated Liver Injury through the EGFR Pathway

The EGFR is a critical target in liver diseases [20]. Through a metabolomics analysis and by investigating the HPA axis, we found that EGFR might be a target for liver injury. The IHC assay showed that liver damage increased the activation of EGFR, and p-EGFR staining showed considerable brown deposits in the rat liver of the anxiety group (Figure 5A). We also performed double IF labeling to determine the effect of p-EGFR on liver tissue marker staining and found that the p-EGFR protein level was considerably higher in the anxiety group (Figure 5B). The semi-quantitative IHC and IF analysis showed a significant increase in positive area (Figure 5C). We found that the ratio of phosphorylated EGFR to total EGFR was higher in stressed rats (Figure 5D,E), although the total EGFR level did not change, which indicated that EGFR was phosphorylated under stress and altered downstream proteins at all levels. At the gene level, the expression of the *EGFR* mRNA was significantly upregulated (Figure 5F). Hence, we investigated its downstream proteins.

The PI3K/AKT and NF-κB pathways were investigated. The ratio of phosphorylated PI3K to total PI3K increased in the PI3K/Akt pathway, along with the ratio of phosphorylated Akt to total Akt (Figure 5G,H), which indicated that the PI3K/AKT pathway was activated by phosphorylation. The NF-κB pathway was triggered initially by the phosphorylation of IKBα (Figure 5G,H). At the gene level, the expression of the *PIK3R1*, *AKT1*, and *NFKBIA* mRNAs was significantly upregulated (Figure 5I). The phosphorylation of IKBα predicts the activation of the NF-κB pathway, which in turn is required for the activation of p65 and p50.

We found that p65 and p50 were phosphorylated. The results of the IHC assay (Figure 6A,C) showed that there were prominent brown deposits in the nucleus of the liver cells of the rats in the anxiety group. We also performed double IF labeling to examine the effects of p-p65 and p-p50 on liver tissue markers. Our results showed that the p-p65 and p-p50 proteins were detected in the nucleus; the expression of p-p50 and p-p65 was elevated in the nucleus of the liver of the rats in the anxiety group (Figure 6B,D). At the protein level, the ratio of p-p65/p65 and p-p50/p50 was significantly enhanced in the stressed rats (Figure 6E,F). At the gene level, the expression of the *NFκB1* and *RELA* mRNAs was significantly upregulated (Figure 6G). We hypothesized that p65 and p50 are involved in nuclear transcription to regulate inflammation. Our findings showed that anxiety-like behavior in rats might be involved in EGFR-mediated inflammatory signaling pathways.

### 2.5. Liver Injury Resulted in the Formation of Inflammatory Factors

We evaluated the changes in inflammatory factors in the liver. At the protein level, TNF, IL-1β, and IL-6 were considerably higher in the stressed rats (Figure 7A,B). Then, we evaluated the levels of TNF, IL-1β, and IL-6 in the peripheral serum and found that these three indicators were significantly elevated in the serum (Figure 7C). This finding indicated that inflammatory factors were produced and released into the peripheral circulation after the liver was damaged due to stress, which confirmed that stress might cause liver inflammation.

## 3. Discussion

Anxiety and depression may cause liver diseases and share some causative agents with cardiovascular disease, implying that psychological distress promotes the occurrence of liver disease [21]. However, only a few studies have used metabolomics to investigate the effects of psychological distress on the liver. 

In this study, the combined stress model was developed. Behavioral testing showed that the frequency and duration of anxious rats entering the open arm both decreased, indicating significantly higher levels of anxiety index in the anxiety group. In the open field experiment, the rats in the anxiety group walked around the box and did not explore the middle area. In contrast, the rats in the normal group explored the middle region more frequently. Therefore, we concluded that the Wistar rats exhibited anxiety-like behaviors. Meanwhile, FST showed that the duration in the anxiety group decreased significantly, indicating that the rats did not exhibit desperate behaviors. The sugar water preference of the rats in the anxiety group decreased significantly in the sugar water preference experiment, indicating that the rats experienced anhedonia after stress. The results of the behavioral test showed that the rats produced anxiety-like behaviors, but the results of the forced swimming test showed that the rats did not experience depression-like behaviors.

Some studies have [22] reported that anxiety-like behaviors may lead to the dysregulation of the HPA axis. When the hypothalamus is stimulated, the pituitary gland releases ACTH, which activates the HPA axis. We found that the ACTH levels increased considerably in the peripheral blood under chronic stress, which indicated activation of the HPA axis. The HPA axis is associated with various stress responses, including anxiety and depression. We found that hyperfunction of the HPA axis in rats was associated with anxiety-like behavior. Also, stress may cause inflammation in the central nervous system or systemically, and the inflammatory environment can trigger several responses that can lead to changes such as anxiety and depression [23]. We also found that the rats experienced neuroinflammation. The results of the behavioral experiments suggested that the rats produced psychiatric-like disorders.

Anxiety-like behavior in rats may cause liver damage. The liver weight and liver index were significantly higher among the rats in the anxiety group. These rats showed inflammatory swelling in the liver, which caused an increase in the liver index. The H&E-stained liver tissue showed that the area of inflammatory infiltration was bigger in the rats of the anxiety group than in those of the control group. Regarding liver function indices, the ALT, AST, and ALP levels were significantly higher in the rats of the anxiety group, indicating liver injury. Several studies have found that psychological distress may promote cardiovascular diseases, suggesting that maintaining a good attitude is essential for a healthy life.

To elucidate the underlying pathological mechanisms, we conducted metabolomics analysis and enrichment analysis, and found that the highest enrichment was in riboflavin metabolism. As the results of the IPA analysis showed that EGFR was a potential target, and differential metabolites closely related to the EGFR receptor were identified, such as choline, Spermidine, and D-Pantothenic acid. EGFR is a widely studied receptor with multiple lipid interactions, and the importance of lipid–protein interactions has received much attention [24,25]. It has been reported in the literature that choline regulates neurogenesis and cortical development by modulating the proliferation and differentiation capacity of neural progenitor cells [26]. In our study, choline was significantly reduced in the anxiety group, suggesting that choline metabolism abnormalities in rats would exacerbate neurological disorders. Spermidine, acting as an NMDA agonist, could modulate memory by promoting NMDAR activity. In the present study, Spermidine was elevated in the liver of anxious rats while there was little change in the brain of the psychological stress model in the reference [27]. We speculated that the increased hepatic Spermidine in rats after stress was blocked since polyamines have limited access to the brain. In another clinical study, it was found that higher biotin intake was associated with lower rates of anxiety and depression. Moderate intake of pantothenic acid was related to a reduced prevalence of anxiety [28]. In addition, D-pantothenic acid was found to be significantly reduced in the anxiety group of rats in our experiment, which is also consistent with the previous research. All these differential metabolites were closely associated with EGFR, indicating that EGFR might be able to serve as a potential target.

EGFR, also known as ErbB1 or HER-1, is a transmembrane receptor belonging to the receptor tyrosine kinase family (RTK) [29]. EGFR consists of an extracellular region that binds ligands, a transmembrane domain, and an intracellular domain which includes a tyrosine kinase domain and a site containing key tyrosine residues at the carboxyl terminus [30,31]. The EGFR system plays a key role in acute and chronic liver disease, and EGFR signaling is hepatoprotective [32]. It is also a precursor to the development of hepatocellular carcinoma [33,34,35]. The etiology of liver disease and cardiovascular disease is partially overlapping, and there is a link between EGFR studies and cardiovascular disease. For example, a study reported that EGFR may be a new therapeutic target for atherosclerosis [36]. EGFR may act as a bridge between liver and cardiovascular disease; thus, psychological factors affecting liver disease may be related to the cardiovascular system. In another study, anxiety-like behavior was found to cause hyperactivity of the HPA axis; the ACTH level was significantly elevated, which was localized to the EGFR receptor [37]. We hypothesized that EGFR may lead to the release of proinflammatory factors in the liver through the stress-induced dysregulation of the HPA axis, eventually resulting in liver inflammation. The risk of cardiovascular disease can increase due to the disruption of the HPA axis [20]. We hypothesized that mood fluctuations may cause the dysregulation of the HPA axis, which may result in cardiovascular disease; these changes can, in turn, lead to liver damage. 

To confirm the results obtained from the metabolomics and neuroendocrine studies, we evaluated the EGFR downstream signaling pathways. After conducting the IHC and IF staining of p-EGFR in the liver tissue, we found a significant difference in the positive staining areas between the anxiety group and the control group. Also, we found that p-EGFR increased significantly while the total protein level remained unchanged. To investigate whether the activation of a downstream protein phosphorylation pathway occurred following EGFR auto-phosphorylation, we conducted a literature review. The research demonstrated that EGFR activation triggered a complex downstream signaling cascade. The primary upstream signaling pathways that were activated included the Ras-Raf-MEK-ERK1/2 pathway and the STAT3 and STAT5 pathways, which also regulate the PI3K-Akt-rapamycin (mTOR) pathway-mediated proliferation and differentiation [38]. Hence, we investigated whether the PI3K/AKT pathway would be activated and lead to the activation of the NF-κB pathway to influence liver inflammation. 

Serine/threonine protein kinase, also known as Akt, participates in numerous processes, such as cell growth, cell death, and glucose metabolism. As Akt is a direct target protein of PI3K, its activation alters several other proteins [39,40]. Transmembrane receptors communicate with tyrosine protein kinases and activate them in response to various stimuli. To determine whether the pathway was activated, we primarily detected the phosphorylation status of the proteins involved in the pathway. Our findings showed that phosphorylation activated the PI3K/Akt pathway. Under conditions of anxiety, this pathway may also be activated and result in inflammation.

The PI3K/Akt pathway promotes NF-κB-dependent transcription, inflammation, immunity, cell proliferation, apoptosis, and several genes associated with physiology and pathology [41,42,43]. Under normal conditions, IKBα and NF-κB p65 and p50 subunits are found in the inactive state in the cytoplasm. Upon activation, IKBα is phosphorylated, leading to ubiquitin-mediated degradation. This process causes the dissociation of p65 and p50, which are further modified and enter the nucleus through the nuclear pore. In this study, the level of p-IKBα, p-p65, and p-p65 proteins in the liver tissue increased considerably, as determined by the IHC and IF detection of p-p65 and p-p65. These results indicated that phosphorylation activates the NF-κB pathway, which controls liver inflammation. The oxidative stress products generated by the activation of the NF-κB pathway and the release of proinflammatory cytokines after the activation of the pathway might, in turn, increase microglia activation and proinflammatory response by stimulating NF-κB, which can continuously exacerbate oxidative stress. The activation of NF-κB and the subsequent inflammatory response can cause cell damage, which in turn exacerbates the process and promotes the development of inflammation. 

To further investigate the production of inflammatory factors by the liver via the NF-κB pathway, we examined the peripheral blood and found that the levels of the inflammatory factors TNF, IL-1β, and IL-6 were significantly increased, and their levels were also significantly higher in the liver, indicating that the liver might promote inflammation and release inflammatory factors into the peripheral blood. Inflammatory factors in the peripheral blood can activate the HPA axis and stimulate the liver, thus causing inflammation in the liver. Therefore, we conducted a preliminary investigation of the mechanism of liver injury induced by anxiety-like behavior in rats. To fully understand the molecular mechanisms involved in this process, the association between the HPA axis and the EGFR receptors and transgenic animal experiments were needed. Considering EGFR is widely distributed in the vascular endothelium, the cardiovascular system might act as a mediator between anxiety disorders and liver disease.

Based on the research that has been reported, the influence of mental illness on physiological health is mostly related through the brain–gut axis. However, the mechanisms associated with the development of liver disease are less elucidated. As far as we know, non-alcoholic fatty liver disease has been found in some research to be highly associated with anxiety, depression, and psychological distress, which interact with each other and have a higher risk of comorbidities. The underlying mechanisms may be connected with HPA axis dysfunction, pro-inflammatory processes, and insulin resistance [44,45]. In the present study, we noted that the HPA axis was dysfunctional in anxiety rats and an inflammatory response in both the brain and the liver was produced, consistent with the mechanisms mentioned above. Certainly, further investigation should be carried out to reveal the core causes of the relationship between psychological distress and liver disease. The summary diagram of our work was presented in Figure 8.

## 4. Methods and Materials

### 4.1. Chemicals and Reagents 

The standards for reserpine (83580–1G) and D4-CA (614149) were purchased from Sigma-Aldrich (St. Louis, MO, USA). Acetonitrile (1.00030.4008; HPLC grade) was purchased from Merck & Co. (Billerica, MA, USA). A Milli-Q system was used to produce ultrapure water (Millipore, MA, USA). The ALP (A059–2–2), ALT (C009–2–1), and AST (C010–2–1) assay kits were purchased from NJJCBIO (Nanjing, China). TNF (MM-0180R2), IL-1β (MM-0047R1), IL-6 (MM-0190R1), ACTH (MM-0565R2), CRH (MM-0520R2), and CORT (MM-0574R2) ELISA kits were purchased from MMBIO (Yancheng, China).

### 4.2. Animals and Grouping 

Specific pathogen-free (SPF) male Wistar rats (*n* = 22, six weeks old, weighing 160–200 g) were purchased from Guangzhou Southern Medical University Co., Ltd. (SCXK [YUE] 2019–0144) (Guangzhou, China). The rats were randomly divided into two groups (n = 11 rats per group), including the anxiety group and the control group. In the anxiety group, rats were housed in solitary cages and were given tail-clamping stimulation and restraint stimulation every day. Tail pinching stimulation was given twice a day for 30 s, respectively, and iodine was used to reduce the inflammation after pinning the tail. The restraint stimulation was given once a day for 3 h by using a restraint cylinder for consecutive 14 d. After grouping, the rats in the anxiety group were kept in isolation (one cage). The rats in the control group were kept in a cage and fed for 14 days without any stimulation. The 11 rats were kept in three cages (*n* = 4, 4, and 3 rats per cage). 

All tests were conducted by the same experimenter. During the trials, the experimenter hid behind a curtain and was not visible from the testing area. In all experiments, the experimenter was blinded while comparing the rats in the two groups. All procedures were conducted following the Guiding Principles in the Care and Use of Animals (China) and were approved by the Laboratory Animal Ethics Committee of Guangzhou University of Chinese Medicine (No. 20220305) on 8 March 2022.

### 4.3. Open Field Test (OFT) 

The XR-SuperMaze animal behavior video monitoring and analysis system (Shanghai Xinruan Information Technology Co., Ltd., Shanghai, China) was equipped with a standard, open, Plexiglas arena (100 cm × 100 cm × 50 cm). Each animal was placed at the center of the apparatus and allotted 10 min to investigate the arena, and then 5 min was selected as the test time. After each test, the arena was thoroughly cleaned with 75% ethyl alcohol. The movement of the animals was tracked using a computer, and the total travel distance and the time spent in the center were evaluated [46,47]. To ensure that anxiety-like behavior was represented, eight rats were used to conduct behavioral assays, which included the open field test, elevated plus maze test, forced swimming test, and sucrose preference test.

### 4.4. Elevated Plus Maze (EPM) Test 

The animal behavior video monitoring and analysis technology (XR-SuperMaze) was used to automatically record data related to the elevated plus maze test (Shanghai Xinruan Information Technology Co., Ltd., Shanghai, China). Initially, the rats were placed in the center of an elevated plus maze, facing an open arm. Then, they were allotted 6 min to explore the maze, and 4 min was selected as the test time [48,49,50]. After each test, the arena was thoroughly cleaned with 75% ethyl alcohol. When two paws of the rats were inside the line designating the entry to a specific arm of the maze, we marked it as the beginning of the time spent in that arm. To assess the locomotor activity of the rats, the number of times they entered each of the arms of the maze was also counted. An anxiety index was used to evaluate the data [51].

### 4.5. Forced Swimming Test (FST)

Using the SuperFst animal behavior video tracking and analysis system, the data on the forced swimming test was automatically recorded (Shanghai Xinruan Information Technology Co., Ltd., Shanghai, China). Each rat was put inside a transparent container (45 cm tall and 20 cm in diameter), containing 15 cm of water at 25 °C. After each test, the water was changed. Each animal was made to swim for 6 min, and then 4 min was selected as the test time. The immobility time was measured when the rats floated in the water without struggling and only moved enough to keep their heads above the water [52]. 

### 4.6. Sucrose Preference Test (SPT)

Each rat was provided two bottles of liquid ad libitum for the duration of the trial, which lasted 24 h. One bottle contained tap water, whereas the other had a solution of 1% sucrose. The location of the bottles was switched after 12 h to avoid any potential effects of the preference for a specific side on the results. Since it was unnecessary to deprive the rats of food and water before the test, they were given access to both ad libitum. The sucrose preference rate was calculated from the ratio of the quantity of sucrose solution consumed to the quantity of total consumption of fluids [53]. 

### 4.7. Sample Preparation for LC-MS

The samples were extracted using a method described in another study [54] with minor modifications. Briefly, the obtained samples were homogenized in cold methanol (MeOH) with the use of a rotor homogenizer. Then, 1 mL of MeOH homogenate was added to 50 mg of liver tissue. The mixture was vortexed for 3 min and centrifuged at 14,000 rpm for 15 min at 4 °C. The supernatant was vacuum dried, reconstituted with internal standards (500 nM reserpine and D4-CA) in 100 µL of MeOH: H_2_O (1:1, *v*:*v*), and centrifuged for 30 min at 4 °C before analysis.

### 4.8. LC-MS Non-Targeted Analysis

We used the 1290 Ultra-High Performance Liquid Chromatograph System (Agilent, Palo Alto, CA, USA) along with the 6540 Q-TOF Mass Spectrometry (Agilent, Palo Alto, CA, USA) for the LC-MS non-targeted analysis; the system was operated in the positive and negative ion modes. The MS settings were as follows: sheath gas temperature of 350 °C, gas temperature of 320 °C, nozzle voltage of 1500 V, a capillary voltage of 3500 V, fragmentor voltage of 175 V, and nebulizer pressure of 35 psi. The flow rates of sheath gas and gas were 11 L/min and 8 L/min, respectively. We used a BEH C18 column (150 mm, 2.1 × 1.7 µm; Waters, Milford, CT, USA) to separate the analytes; the column temperature and flow rate were maintained at 50 °C and 0.4 mL/min. The LC conditions were as follows: Solvent A comprised 0.1% formic acid in deionized water, and solvent B comprised 0.1% formic acid in acetonitrile. The gradient elution was initiated with 2% B for 1 min, linearly increased to 100% B in 15 min, and maintained at 100% B for 3 min. After 1 min of re-equilibration, the composition was returned to its original ratio of 2% B.

### 4.9. Identification of Metabolites and Pathway Analysis

The Mass Hunter Workstation software (version B.07.00; Agilent, Palo Alto, CA, USA) was used to collect raw data in both ionization modes. To obtain multivariate data of each sample, such as all characteristics of ions, RT, *m*/*z*, and intensity, the raw data format extracted using Profinder 10.0 (Agilent, Palo Alto, CA, USA) and Mass Profiler Professional 15.0 (Agilent, Palo Alto, CA, USA) was used for peak selection, alignment, and integration. The data were imported into SIMCA-P (version 14.0; Umetrics, Ume, Sweden) for conducting multivariate statistical analyses, which included principal component analysis (PCA) and orthogonal partial least squares discriminant analysis (OPLS-DA). The results were used to determine the VIP value for each feature. The control and anxiety groups were compared using the Student’s *t*-test or Mann–Whitney U test. The features were considered to be potentially differential metabolites if they followed the conditions of VIP > 1 and *p* < 0.05. The data obtained from the Metlin, the Human Metabolome Database (HMDB), and those obtained from our laboratory database were compared with accurate mass data and isotopic profiles of compounds and their product ions. Pathway analysis was performed using Metaboanalyst 5.0. For the pathway enrichment analysis, potential targets were found using the IPA software (version 1.0; QIAGEN, DUS, Germany).

### 4.10. Hematoxylin and Eosin Staining (H&E), Immunohistochemistry (IHC), and Immunofluorescence (IF)

After the rats were sacrificed, their liver tissue was removed and first fixed in 4% paraformaldehyde and then dehydrated and embedded in paraffin. An RM2016 microtome (Leica, Shanghai, China) was used to cut 4 µm slices, which were stained with hematoxylin and eosin (H&E). 

Immunohistochemistry assay was performed following the method described in another study [55]; p-EGFR antibodies (Cell Signaling Technology, #3777), p-p50 antibodies (Affinity, AF3219), and p-p65 antibodies (Affinity, AF2006) were used to perform the assay. Antibodies should be diluted (1:400) with buffer solutions that stabilize the antibodies for IHC staining. The section was blocked and incubated for 10–30 min at room temperature. The blocking buffer was decanted, the primary antibody was added, and the membrane was incubated for 1 h with shaking. After that, the membrane was washed and the HRP-conjugated secondary antibody was added for 1 h at room temperature with shaking. Then, the hydrogen peroxide was added to the DAB solution and was incubated until the desired development was achieved. The positive areas were specifically stained brown.

The images were viewed under a light microscope (E100; Nikon Corporation, Tokyo, Japan) and a fluorescence microscope (DMI3000 B; Leica, Wetzlar, Germany). A semi-quantitative analysis was conducted with eight images using the Image J software (v1.51). The details are as follows.

First, H&E images were imported into Image J and manually framed using rectangles, circles, and polygons. The actual size of the images was adjusted according to the scale, and then the area of inflammatory infiltration was measured.

The IHC images were imported into the IHC Toolbox plug-in of the Image J software (v1.51) for processing. After setting the image type (8 bit), the threshold for IHC positive area (Image-Adjust-Threshold-Auto) and the parameters (Analyze-Set Measurements-Area, Limit to threshold) were set. Finally, the measurements were made (Analyze-Measure).

For analyzing IF images, the single channel (Image-Color-Split Channels) was extracted, the threshold value was adjusted, and the appropriate area (Image-Adjust-Threshold) was selected. Then, “Analyze-Set Measurements” was selected and the parameters to be measured were chosen. The “Limit to Threshold” option should also be selected, because otherwise the entire image will be measured, rather than the selected area. Next, the results table appeared by selecting “Analyze-Measure”, which was saved.

### 4.11. Western Blotting Assay

We evaluated the expression of rat liver proteins from six rats in each group; 50 mg liver tissue was taken for protein isolation in each rat. We prepared the tissue lysates in RIPA buffer and performed Western immunoblotting. Equal amounts of liver protein extract (50 μg/10 μL) were processed via 10% SDS PAGE to separate the proteins. After transferring the proteins onto a PVDF membrane, the membrane was washed and incubated with 5% BSA in TBST buffer for 2 h at room temperature. After washing twice with TBST, the membrane was incubated with the primary antibodies overnight at 4 °C. Then, the samples were incubated with horseradish peroxidase-conjugated secondary antibodies for 2 h. Finally, using enhanced chemiluminescence (ECL), the chemiluminescence signals were detected, and the Image J software (v1.51) was used to quantitate the bands. The antibodies used in this study are presented in Appendix A; we confirmed the specificity of the antibodies. Each target protein had a corresponding loading control on the same membrane.

### 4.12. Real-Time PCR Analysis

First, total RNA was extracted from 20 mg liver samples of each rat (five rats per group) using TRIzol reagent (Accurate Biotechnology Co., Ltd., Changsha, China) following the manufacturer’s protocol. First-strand cDNA was generated using 1 µg of total RNA with an SYBR Green I reagent kit (Accurate Biotechnology Co., Ltd., Changsha, China). The real-time PCR reaction was performed using a QuantStudioTM 5 system (Applied Biosystems, Stony Creek, VA, USA) and the QuantStudioTM design and analysis software (v.1.5.1). All primers were designed according to the Primer 5.0 website and purchased from Sangon Biotech (Shanghai, China), which were listed in Appendix A. The target mRNA was normalized to GAPDH; GAPDH served as an endogenous control. The changes in the relative expression were determined using the 2^−ΔΔCt^ method.

### 4.13. Statistical Analysis

We determined the sample size by using The Power and Sample Size Website (http://powerandsamplesize.com/, accessed on 13 April 2023) and calculated the sample size based on the anxiety index metric. The average anxiety index of the normal group was 79.69%, and that of the anxiety group was 82.82%. The sample size was calculated to be seven (standard deviation = 2, Power = 0.80, Type I error rate = 5%). In this study, 11 rats were used in the experiments, which met the sample size requirement.

For normally distributed and non-normally distributed variables, univariate analysis was performed based on the Student’s *t*-test or Mann–Whitney U test using the SPSS v22.0 software (Chicago, IL, USA). We evaluated the normality and homogeneity of variance as follows. In the normality test, when the *p*-value was above 0.05, the data were considered to follow a normal distribution. Regarding the homogeneity of variance, when the *p*-value was above 0.05, the assumption of the homogeneity of variance was met. To determine the VIP value, multivariate analyses were conducted based on PCA and OPLS-DA using the SIMCA-P 14.0 software (Umetrics, Ume, Sweden). All images were made using GraphPad Prism v6 (San Diego, CA, USA). The statistical analysis is presented in the Appendix A.

## 5. Conclusions

In this study, we showed that the EGFR-related signaling pathways and the HPA axis might be involved in the development of liver injury caused by anxiety-like behavior in rats. The results of this study provided novel insights into psychiatric diseases and anxiety disorders. The cardiovascular system may act as a link between liver damage and anxiety, and these findings provide new ideas for future studies.

## Figures and Tables

**Figure 1 ijms-24-13356-f001:**
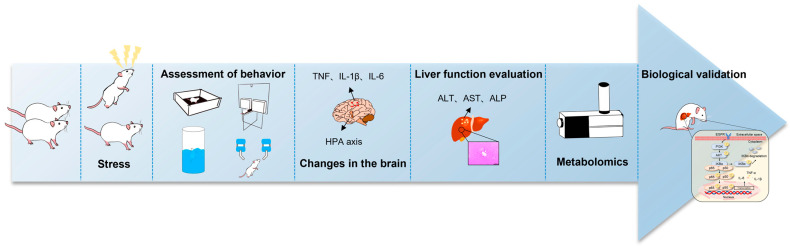
A flow chart of the study.

**Figure 2 ijms-24-13356-f002:**
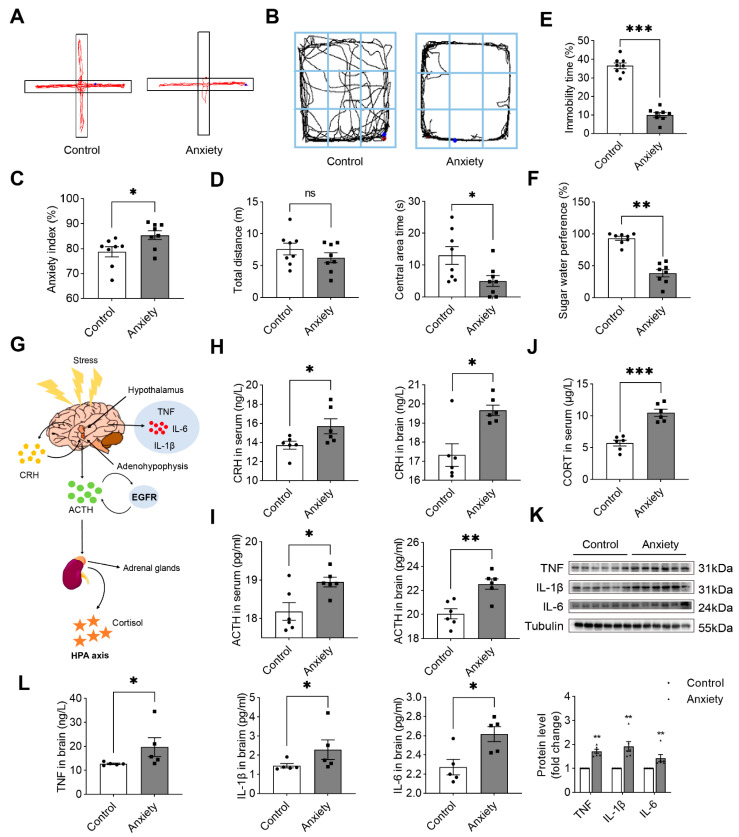
Detection of maternal behavior. (**A**) Action trajectory diagram of rats in EPM. (**B**) Action trajectory of rats in OFT. (**C**) The percentage of anxiety index was determined by the elevated plus maze test (*n =* 8). (**D**) The total distance moved and the time spent in the central area by rats in 5 min were measured and recorded as the OFT score (*n =* 8). (**E**) The percentage of forced swim tests showing immobility time (*n =* 8). (**F**) The percentage of sucrose solution consumed was calculated to determine sucrose preference in the experiment (*n =* 8). (**G**) A diagram illustrating the mechanism involving the HPA axis. (**H**) Changes in corticotropin-releasing hormone in the blood and brain (*n* = 6). (**I**) Changes in adrenocorticotropic hormone in the blood and brain (*n* = 6). (**J**) Changes in corticosterone in the blood (*n* = 6). (**K**) Western blot analysis was performed on liver tissue lysates of rats in the control and anxiety groups, and the blots were quantified (*n* = 6). (**L**) Changes in TNF, IL-1β, and IL-6 in the brain (*n* = 5) versus the control; * *p* < 0.05, ** *p* < 0.01, and *** *p* < 0.001. ns, no significance. The data are expressed as SEM.

**Figure 3 ijms-24-13356-f003:**
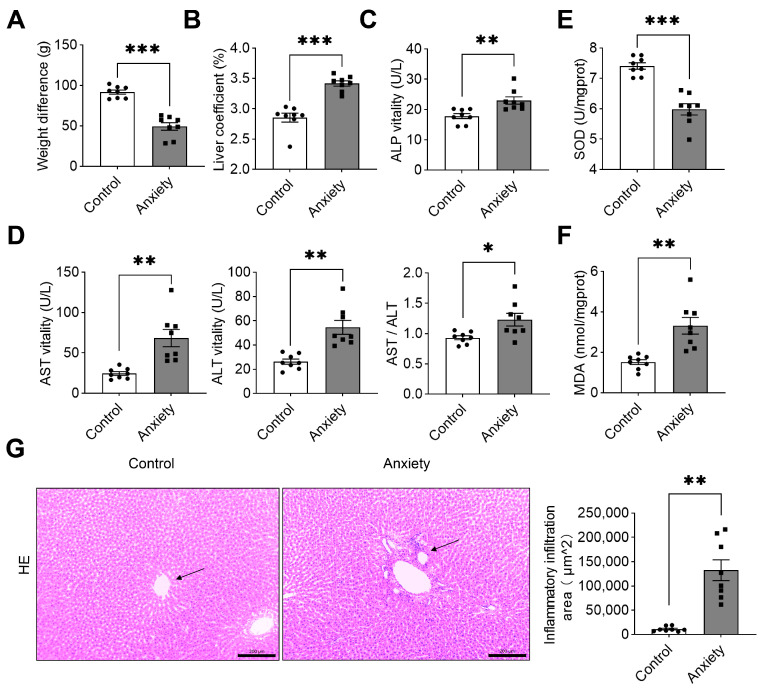
Anxiety-related stress resulted in liver injury. (**A**) Changes in body weight before and after the experiment (*n =* 8). (**B**) The isolated liver of the rats was weighed, and the liver coefficient was calculated (*n =* 8). (**C**,**D**) Serum levels of ALP, AST, ALT, and the ratio of AST/ALT (*n =* 8). (**E**,**F**) SOD and MDA level in rat liver tissue (*n =* 8). (**G**) Representative H&E staining of liver tissue (magnification: 100×) (*n =* 8); * *p* < 0.05, ** *p* < 0.01, and *** *p* < 0.001 versus the control. The data are expressed as SEM.

**Figure 4 ijms-24-13356-f004:**
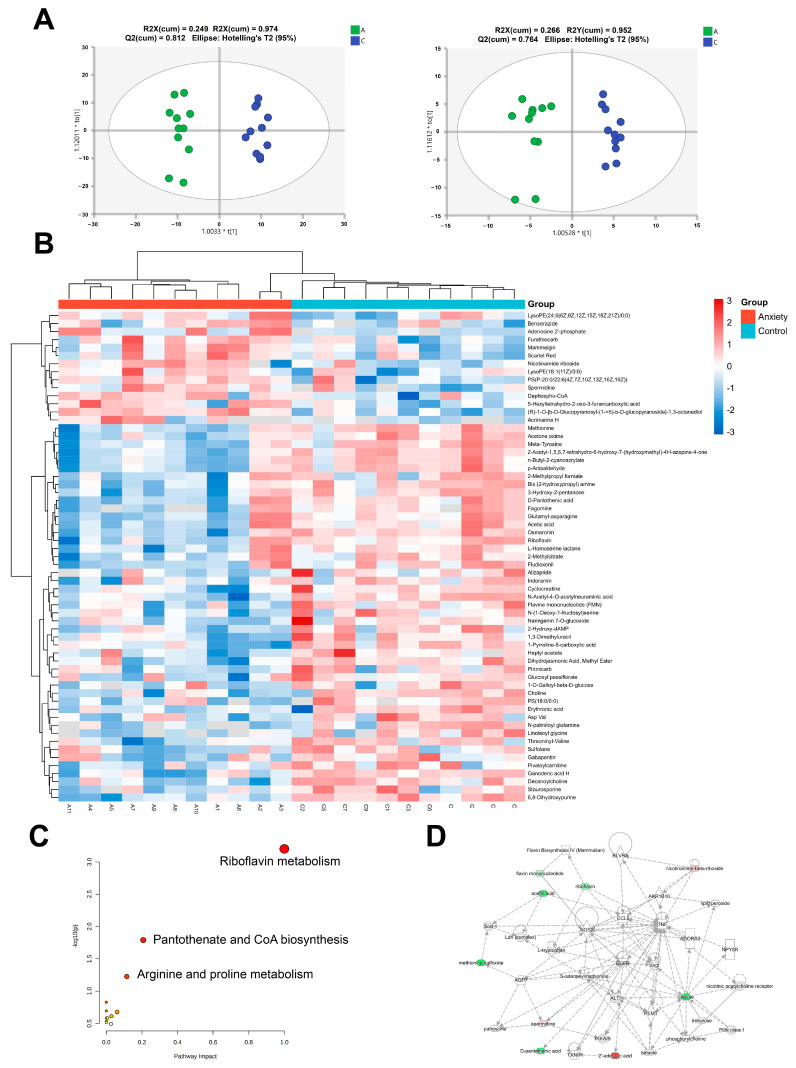
Overview of liver metabolomics in the study. (**A**) The OPLS-DA scatter plot of the metabolic profiles in positive ion mode and negative ion mode. (**B**) The heat map analysis of 61 differential metabolites. (**C**) Differential metabolite pathway analysis. (**D**) The relationship between the potential targets and the EGFR pathway, as determined via the IPA network analysis. The data are expressed as SEM.

**Figure 5 ijms-24-13356-f005:**
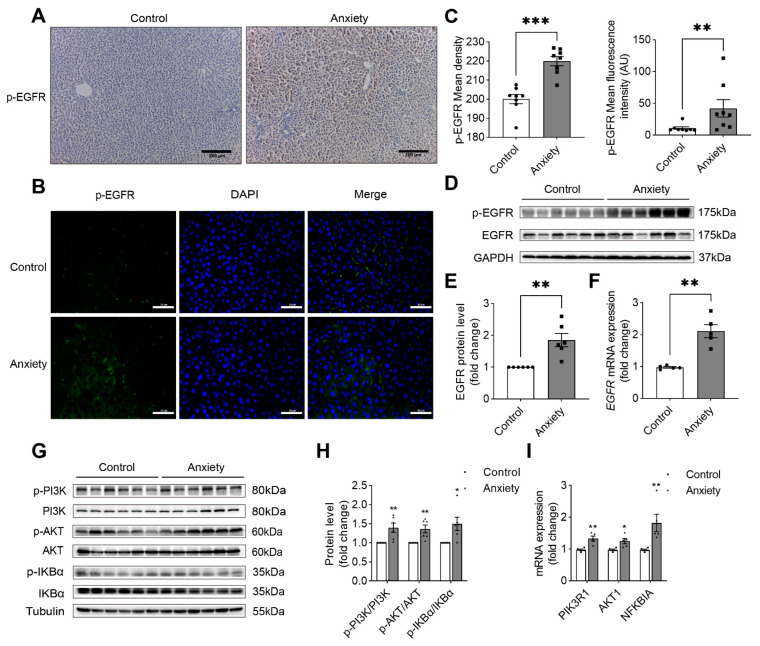
Validation of the proteins in the EGFR/PI3K/AKT pathway. (**A**) Representative micrographs of liver sections of rats in the control and anxiety groups after the sections were stained with phospho-EGFR immunohistochemistry. (**B**) Representative micrographs of liver sections of rats in the control and anxiety groups after the sections were stained with phospho-EGFR immunofluorescence. (**C**) Quantification of IHC and IF (*n* = 8). (**D**) The Western blotting analysis of liver tissue lysates of rats in the control and anxiety groups. (**E**) Quantification of the blots (*n* = 6). (**F**) The real-time PCR analysis of the expression of the *EGFR* mRNA. (**G**) The Western blotting analysis of liver tissue lysates of rats in the control and anxiety groups. (**H**) Quantification of the blots (*n* = 6). (**I**) The real-time PCR analysis of the expression of the *PIK3R1*, *AKT1*, and *NFKBIA* mRNAs (*n* = 5) versus the control; * *p* < 0.05, ** *p* < 0.01, and *** *p* < 0.001. The data are expressed as SEM.

**Figure 6 ijms-24-13356-f006:**
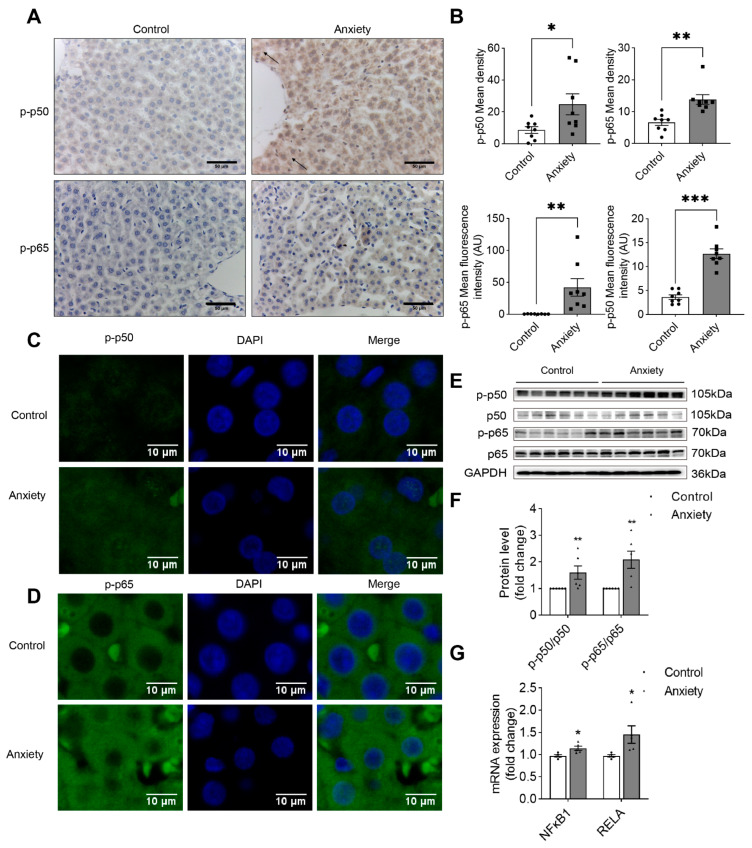
Validation of the proteins in the NF-κB pathway. (**A**) Representative micrographs of liver sections of the rats in the control and anxiety groups after the sections were stained with phospho-NF-κB p65 and phospho-NF-κB p50 for immunohistochemistry. (**B**) Quantification of IHC (*n* = 8). (**C**) Representative micrographs of the liver sections of the rats in the control and anxiety groups after the sections were stained with phospho-NF-κB p65 and phospho-NF-κB p50 IF. (**D**) Quantification of IF (*n* = 8). (**E**) The Western blotting analysis of liver tissue lysates of the rats in the control and anxiety groups. (**F**) Quantification of the blots (*n* = 6). (**G**) The real-time PCR analysis of the expression of the *NFκB1* and *RELA* mRNAs (*n* = 5) versus the control; * *p* < 0.05, ** *p* < 0.01, and *** *p* < 0.001. The data are expressed as SEM.

**Figure 7 ijms-24-13356-f007:**
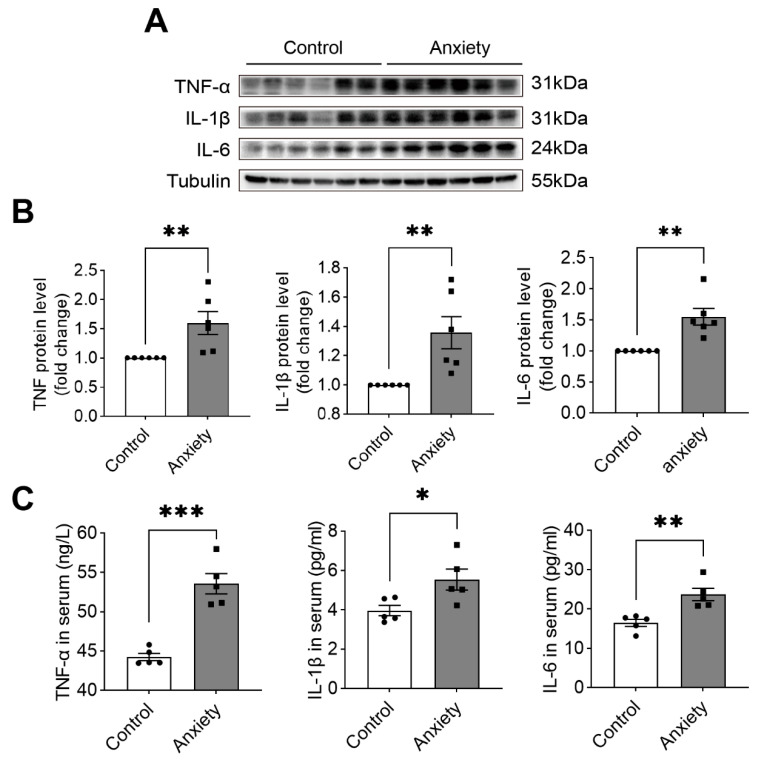
Detection of inflammatory factors. (**A**) The Western blotting analysis of liver tissue lysates of the rats in the control and anxiety groups. (**B**) Quantification of the blots (*n* = 6). (**C**) Serum levels (*n* = 5) of TNF, IL-1β, and IL-6 versus the control; * *p* < 0.05, ** *p* < 0.01, and *** *p* < 0.001. The data are expressed as SEM.

**Figure 8 ijms-24-13356-f008:**
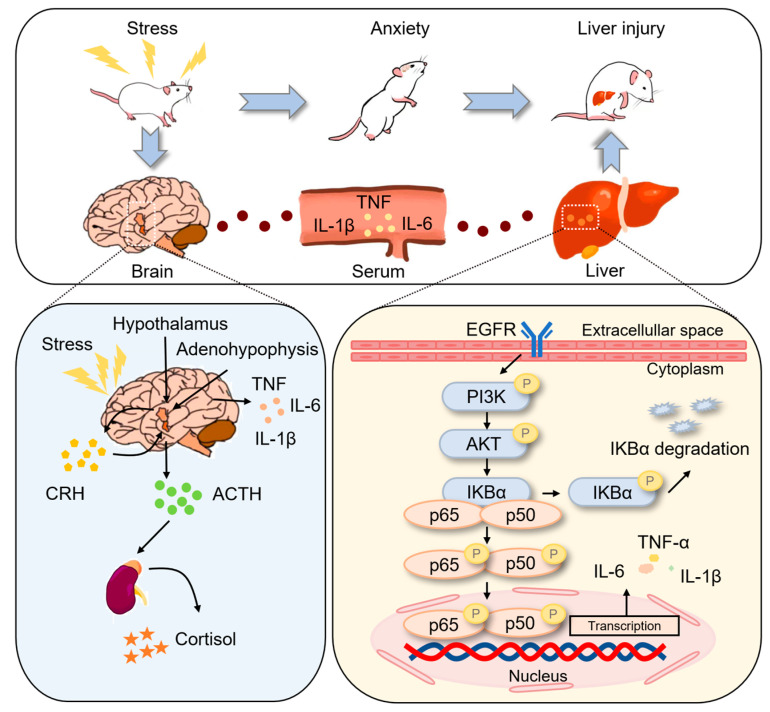
Summary diagram.

## Data Availability

All relevant data were within the paper and its Appendix A.

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
