# Peer review of "Unravelling the Link between Psychological Distress and Liver Disease: Insights from an Anxiety-like Rat Model and Metabolomics Analysis"

_ijms, 2023, doi:10.3390/ijms241713356_

Round 1

Reviewer 1 Report

This paper discusses a study that explores the link between psychological distress and liver disease. The study uses an anxiety-like rat model and metabolomics analysis to investigate this connection. The researchers screened metabolites to understand the etiology of stress-induced liver injury better. They found and analyzed differential metabolites using a heat map and differential metabolite pathway analysis. They also used IPA network analysis to determine the relationship between potential targets and the EGFR pathway. The study found specific metabolites associated with psychological distress and liver disease. The findings could potentially be applied in the development of new treatments for liver disease.

There are a few things I'd like to suggest:

1. Using a rat model and metabolomics analysis, the study aims to investigate the relationship between psychological distress and liver disease. The study aims to contribute to the understanding of the molecular mechanisms involved in the relationship between psychological distress and liver disease, which could potentially lead to the development of new treatments for liver disease associated with psychological distress. Therefore, the problem that this study aims to address is the lack of understanding of the molecular mechanisms involved in the relationship between psychological distress and liver disease.

2. Using metabolomics analysis, the study found different metabolites associated with psychological distress and liver disease. However, the specific metabolites were not mentioned in the literature.

3. The results of this study could potentially be used to develop new treatments for liver disease. The study found that anxiety-like behavior caused liver damage to some extent and was closely associated with the EGFR-related pathway. Therefore, targeting this pathway could be a potential strategy for treating liver disease associated with psychological distress. However, more research is needed to fully understand the molecular mechanisms involved and to develop effective treatments.

4. There are limitations to the rat model used in the study that may affect the generalizability of the results. The study used an anxiety-like rat model to investigate the relationship between psychological distress and liver disease. However, it is important to note that animal models may not fully represent the complexity of human psychological distress. In addition, the study used a small sample size, which may limit the generalizability of the results. Therefore, further research is needed to confirm the findings of this study and to determine the applicability of the results to humans.

Minor editing of English language required

Reviewer 2 Report

This manuscript prepared by Liu and coauthors, aims to study the effect of psychological distress on liver disease.  The text can be potentially interesting for the readers, since recently there is growing evidence about the effect  psychological distresses on human body health. Despite this fact, the manuscript needs to be improved before being accepted for publication. If possible, the answers and comments should be also included in a new version of manuscript, since they will allow better understanding of presented data.

Line 150 – what is MeOH homogenate?

Line 190 – since the manuscript is easy to follow ad details of immunohistochemistry assay; add the dilutions od antibodies used; what was the method for detection? (i.e., secondary antibody conjugated with…)

Line 208- what was the “selected threshold algorithm”?

Line 213 -   please confirm that you used Laemmli sample buffer to obtain protein extracts; what was the concentration of protein loaded during SDS-PAGE?

Does one liver taken from rat is enough to obtain sufficiently isolation of protein, RNA, and then perform ICH? What was the mas of liver taken for protein isolation? What was the mass of liver taken for RNA isolation? What is the source for starters’ sequences? Present the technical details of each of the methods used.

For how long the animals were treated to obtain the “anxiety group”? Present the details of method used for induction of anxiety. Haw it differs from other used protocols?

Have the Authors checked in blood any other parameter, such as oxidative stress? These results will greatly enriched the manuscript.

Enlarge all the photos of cells. Scale bar is almost not ready to be read.

Please explain why all wb results obtained for control are not equal to 1.

Figure 4 – obligatory enlarge the heat map.

In summary, the manuscript requires at least the major revision.

Round 2

Reviewer 2 Report

I have read the Authors’ response and a new version of text; they answered most my questions and they improved the manuscript. In my opinion the text can be published in IJMS journal.